# "*Candidatus* Campylobacter infans" detection is not associated with diarrhea in children under the age of 2 in Peru

**Paul F. Garcia Bardales**[1], **Francesca Schiaffino**[2,3], **Steven Huynh**[4], **Maribel Paredes Olortegui**[1], **Pablo Peñataro Yori**[1,2], **Tackeshy Pinedo Vasquez**[1], **Katia Manzanares Villanueva**[1], **Greisi E. Curico Huansi**[1], **Wagner V. Shapiama Lopez**[1], **Kerry K. Cooper**[5]*, **Craig T. Parker**[4], **Margaret N. Kosek** [1,2]*

**1** Asociacion Benefica Prisma, Iquitos, Peru, **2** Division of Infectious Diseases, University of Virginia, Charlottesville, Virginia, United States of America, **3** Faculty of Veterinary Medicine, Universidad Peruana Cayetano Heredia, San Martin de Porres, Lima, Peru, **4** Agricultural Research Service, U.S. Department of Agriculture, Produce Safety and Microbiology Research Unit, Albany, California, United States of America, **5** School of Animal and Comparative Biomedical Sciences, University of Arizona, Tucson, Arizona, United States of America

* kcooper@arizona.edu (KKC); mkosek@virginia.edu (MNKI)

**Data Availability Statement:** Short read data are available at NCBI SRA and are associated with

## Abstract

A working hypothesis is that less common species of *Campylobacter* (other than *C. jejuni* and *C. coli)* play a role in enteric disease among children in low resource settings and explain the gap between the detection of *Campylobacter* using culture and culture independent methods. "*Candidatus* Campylobacter infans" *(C. infans)*, was recently detected in stool samples from children and hypothesized to play a role in *Campylobacter* epidemiology in low- and middle-income countries (LMIC). This study determined the prevalence of *C. infans* in symptomatic and asymptomatic stool samples from children living in Iquitos, Peru. Stool samples from 215 children with diarrhea and 50 stool samples from children without diarrhea under the age of two were evaluated using a multiplex qPCR assay to detect *Campylobacter* spp. (16S rRNA), *Campylobacter jejuni / Campylobacter coli* (*cadF* gene), *C. infans* (*lpxA*), and *Shigella* spp. (*ipaH*). *C. infans* was detected in 7.9% (17/215) symptomatic samples and 4.0% (2/50) asymptomatic samples. The association between diarrhea and the presence of these targets was evaluated using univariate logistic regressions. *C. infans* was not associated with diarrhea. Fifty-one percent (75/146) of *Campylobacter* positive fecal samples were negative for *C. jejuni*, *C. coli*, and *C. infans* via qPCR. Shotgun metagenomics confirmed the presence of *C. infans* among 13 out of 14 positive *C. infans* positive stool samples. *C infans* explained only 20.7% of the diagnostic gap in stools from children with diarrhea and 16.7% of the gap in children without diarrhea. We posit that poor *cadF* primer performance better explains the observed gap than the prevalence of atypical non-C. jejuni/coli species.

BioProject PRJNA834762. All other relveant data are within the manuscript and its supporting information files.

**Funding:** Funding for this study was provided by the Bill and Melinda Gates Foundation (OPP1066146 and OPP1152146 to MNK), the National Institutes of Health of the United States (R01AI158576 and R21AI163801 to MNK and CP; D43TW010913 to MNK). This research was also supported in part by USDA-ARS CRIS project 2030-42000-055-00D ( to CP). The funders had no role in study design, data collection and analysis, decision to publish, or preparation of the manuscript.

**Competing interests:** NO authors have competing interests.

## Author summary

A potentially new species of *Campylobacter*, "*Candidatus* Campylobacter infans" has been recently identified using culture independent diagnostic techniques. This study determined its prevalence using qPCR in diarrhea and non-diarrheal samples of children under the age of two, and confirmed results using shotgun metagenomics. The presence of "*Candidatus* Campylobacter infans" was not associated with diarrhea in this population.

## Introduction

Campylobacteriosis is a leading cause of diarrhea, enteropathy and stunting among children living in poverty [1–3]. The isolation and genomic characterization of these pathogens are limited in low- and middle-income countries (LMIC) where the burden of disease is highest. This is partly a result of a larger proportion of cases diagnosed using nucleic acid-based techniques or ELISA, which is higher in low-income settings in comparison to high-income settings and in part secondary to limited capacity and funds to sequence genomes that are isolated in LMIC settings [4,5]. The diagnostic deficiency of culturing is likely due to imperfect sensitivity of traditional culture methods, even for common *Campylobacter* species such as *C. jejuni* and *C. coli* [6,7]. Additionally, uncommon species of *Campylobacter* that have been shown to be causative agents of enteric disease among children in low resource settings are diverse and fastidious in nature, requiring special atmospheric gas mixture and nutrient requirements for culturing. These uncommon species also require the creation and validation of new assays for nucleic acid-based methods. The material for both culture and nucleic acid-based methods are especially difficult to obtain in LMIC [8–11]. Among uncommon *Campylobacter* species, *Campylobacter upsaliensis*, *Campylobacter hyointestinalis* and a potential new species of *Campylobacter*, "*Candidatus* Campylobacter infans" *(C. infans)*, have been identified in stool samples from children and hypothesized to play a role in the epidemiology of *Campylobacter* infection within highly endemic communities [8,9,12].

 *Campylobacter infans* was first described in fecal samples of symptomatic and asymptomatic children under 1 year of age enrolled in the Global Enteric Multicenter Study (GEMS) [13]. This potential new species was identified from a single fecal sample associated with an infant experiencing prolonged diarrhea where *Campylobacter* spp. represented 83% of the fecal microbiome. Through assembly of shotgun metagenomic sequencing reads from this fecal sample, 75 contigs consisting of ~1.7 Mb were described as *C. infans*, which showed less than 75% similarity to the genomes of all other recognized *Campylobacter* species [12]. *C. infans* has been isolated once in a patient with chronic diarrhea living with HIV in Europe. In this instance, the detection of *C. infans* was not clearly associated with the chronic diarrhea reported, yet it represents the only report of its isolation [14,15]. Further query using both *lpxA* and *atpA* markers specific for *C. infans* in samples derived from the GEMS cohort identified the potential new *Campylobacter* species in 10% of fecal samples analyzed, all of which were positive for *Campylobacter* spp. by molecular diagnostics [12]. Because of this bias, the prevalence of this novel species in cases of diarrhea and in stools from asymptomatic children in a LMIC study, independent of *Campylobacter* status, has yet to be characterized.

 The objective of this study was to compare the prevalence of *C. infans* in stool samples from children with acute diarrheal disease to stool samples from asymptomatic children living in a low resource community with established high rates of *Campylobacter* disease. Additionally, we set to determine if the presence of *C. infans* in these stool samples would explain the high discrepancy of the detection of *Campylobacter* between nucleic acid diagnostics and culture.

## Methods

### Ethics statement

Samples used in this study were collected as part of studies approved by the Institutional Review Boards of Asociacion Benefica Prisma (Lima, Peru) and Johns Hopkins Bloomberg School of Public Health. Parents or legal guardians of the participants of both studies provided a written consent to participate in the research and consented for future use of biological specimens that fell within the scope of this project.

### Biological samples

Fecal samples were derived from children with diarrhea under two years of age seeking health care at local primary and tertiary care centers in Iquitos, Loreto, Peru. Stool samples from children who had not had diarrhea in the month prior to enrollment in a study of biomarkers of environmental enteropathy were analyzed as a population-based reference sample. This reference population lies within the population catchment area of recruitment for the study of acute diarrhea. All stools were collected between 2018–2021. Samples were collected from both groups of participants upon enrollment and stored at -70˚C from case and reference populations.

To assess the hypothesis that *C. infans* may also be an oral bacterium, we evaluated its presence in saliva samples from children who had *C. infans* positive and *C. infans* negative fecal samples. Saliva samples from a random subset of the same participants were collected using Oracol S10 swabs (Malvern Medical Developments, Worcester, UK). Saliva samples were heated at 60˚C for 30 minutes, centrifuged at 4000 rpm for 10 minutes and 500 μL of the supernatant were stored at -70˚C until further processing

### DNA extraction and qPCR

Fecal DNA was extracted from 0.2 grams of feces using the QIAamp DNA Stool Mini Kit (Qiagen, Carlsbad, CA), according to the manufacturer's instructions. DNA was extracted from 200 μL of saliva using PureLink Genomic DNA mini kit (ThermoFisher Scientific, Massachusetts), according to the manufacturer's instructions.

A negative control consisting of RNA and DNA free water was used for each extraction set. All samples were processed using a Taqman based multiplex assay to detect *Campylobacter* spp. (16S rRNA gene), *Campylobacter jejuni* / *Campylobacter coli* (*cadF* gene), *C. infans* (*lpxA* gene) and *Shigella* spp. (*ipaH* gene), using the primers and probes specified in **Table 1.** The final assay consisted of a 25 μL final reaction mixture with 12.5 μL of Environmental Master Mix (2X) (Applied Biosystems, Foster City, CA), forward and reverse primers (0.2 μM), probes (0.1 μM), 1 μL of DNA template and RNase and DNase free water (Ambion, Thermo Fisher Scientific, Waltham, MA, USA). The assays were performed on a QuantStudio 7 Flex (Applied Biosystems, Foster City, CA) using the following cycling conditions: 95˚C for 10 minutes followed by 45 cycles of 95˚C for 15 seconds and 60˚C for 1 minutes. Custom manufactured double-stranded synthetic DNA fragments (gBlocks, Integrated DNA Technologies, Coralville, IA, USA) were used as positive controls (**S1 Table**). Negative template controls (RNase and DNase free water) were included in each amplification reaction. For quality control purposes, 10% of samples were run in duplicate.

Standard curves for each marker were prepared using 10-fold serial dilutions of synthetic positive controls (6.0 x 10$^4$–6.0 x 10˚) gene copies/μL. A cut-off cycle threshold of 38 was used to determine positivity based on the assays limit of detection. Gene copy number of each qPCR marker per gram of feces were calculated and log(10) transformed for all fecal samples.

**Table 1. Primers and probes for the detection of *Campylobacter* spp., *Campylobacter jejuni/coli*, *candidatus* Campylobacter infans and *Shigella* spp.**

| Target | Nombre | Sequence | Source |
|---|---|---|---|
| *Campylobacter* genus (16S rRNA) | 16s_Fw | 5'- CAC GTG CTA CAA TGG CAT AT -3 | [4] |
| | 16s_Rv | 5'- GGC TTC ATG CTC TCG AGT T -3' | |
| | 16s_Probe | 5'- /56-FAM/CAG AGA ACA /ZEN/ ATC CGA ACT GGG ACA /3IABkFQ/ -3' | |
| *Campylobacter coli* and *Campylobacter jejuni* (*cadF*) | cadF_Fw | 5'- CTG CTA AAC CAT AGA AAT AAA ATT TCT CAC -3' | [4] |
| | cadF_Rv | 5'- CTT TGA AGG TAA TTT AGA TAT GGA TAA TCG -3' | |
| | cadF_Probe | 5' -/56-VIC/CAT TTT GAC /ZEN/ GAT TTT TGG CTT GA/3IABkFQ/ -3' | |
| *Shigella* spp. (*ipaH*) | ipaH_Fw | 5'- CCT TTT CCG CGT TCC TTG A -3' | [28] |
| | ipaH_Rv | 5'- CGG AAT CCG GAG GTA TTG C -3' | |
| | ipaH_Probe | 5'- /56-TAMN/CGC CTT TCC GAT ACC GTC TCT GCA/3IAbRQSp/ -3' | |
| *Candidatus Campylobacter infans* (*lpxA*) | Infans_Fw | GGC CAT TAT AAA AGC ATT ATC GCC TA | This study |
| | Infans_Rv | CTT TGT GGG TAA AAA TGC CGT AAT TG | |
| | Infans_Probe | 3- /5Cy5/CGT GCC TGA/TAO/GTT TAT GCA AAA CTC C/3IAbRQSp/ | |

Given that there are three copies of the 16S rRNA gene present in each *Campylobacter* genome, gene copy numbers for this target were adjusted accordingly. A single copy of the *cadF* gene (*C.jejuni/ C.coli*) or the *lxpA* gene (*C. infans*) is present per genome making this adjustment unnecessary for these gene targets.

## Statistical analysis

Binary variables indicating the presence and absence of each qPCR target were created based on the established cycle threshold cut-off. A sample was positive if the detected cycle threshold was below the limit of detection. The presence and absence of *Campylobacter* spp., *C. jejuni / C. coli*, *C. infans* and *Shigella* spp. was tabulated. The association between diarrhea and the presence of *Campylobacter* spp., *C. jejuni / C. coli*, *C. infans*, and *Shigella* spp. was evaluated using univariate logistic regressions with 95% confidence intervals. The difference in the mean log(10) gene copy number per gram of symptomatic and asymptomatic fecal sample was assessed using an independent two-sample T-test. Statistical analysis and data visualization was performed in STATA 14 and R (version 4.1.1).

## Shotgun metagenomic confirmation of *C. infans*

**Sequencing of metagenomic DNA.** Shotgun metagenomics was conducted on DNA from a subset of fecal samples with positive qPCR signals for *C. infans* by sequencing on an Illumina MiSeq sequencer. The libraries were prepared using Illumina DNA Prep Tagmentation kit (Illumina, San Diego, CA), following the manufacturer's instructions except for changes that increased library insert size to a median of 631 bp and a size range between ~375–1100 bp. This was done by decreasing the 1st and 2nd volume of Sample Purification Beads to 40 µl and 11 µl, respectively. The increase was necessary since standard manufacturer's instructions resulted in inserts mostly below 300 bp. Libraries had a final elution of 10 µl Illumina resuspension buffer. Illumina-DNA/RNA UD Indexes Plate A, B, C and D dual index adapters were ordered from Integrated DNA Technologies (Coralville, IA) and used at 1 µM final concentration. Instead of pooling equal volume, individual libraries were quantified using the KAPA Library Quantification Kit (Roche), since we found qPCR to be a more accurate quantification than using equal volume. Libraries were quantified in 10 µl volume reactions and 90-s annealing/extension PCR, and then pooled and normalized to 4nM. Pooled libraries were re-quantified by ddPCR on a QX200 system (Bio-Rad, Hercules, CA), using the

Illumina TruSeq ddPCR Library Quantification Kit and following the manufacturer's protocols. Libraries were sequenced using a 2 x 250 bp paired end v2 reagent kit on a MiSeq instrument (Illumina) at 16 pM, following the manufacturer's protocols. Short read data are available at NCBI SRA and are associated with BioProject PRJNA837236.

**Detection of *C. infans*.** Detection of *C. infans* was performed using the reference assembler within Geneious Prime v2021.2.2 (Biomatters, Ltd., Auckland, New Zealand). Illumina paired sequence reads from fourteen metagenomic samples with the number of reads per sample ranging from 392,074 to 10,720,574 reads were assembled using the Geneious Mapper with low sensitivity settings (<10% mismatch between read and reference) to *"Candidatus* Campylobacter infans" strain 19S00001 chromosome (CP049075.1). The number of reads assembled from the different samples ranged from 0 and 68,948.

***Detection of other* campylobacter *species*.** Detection of *Campylobacter* species was also performed using the reference assembler within Geneious Prime v2021.2.2 (Biomatters, Ltd., Auckland, New Zealand). Illumina paired-end reads >100 nt generated from an individual fecal sample were simultaneously mapped to 28 *Campylobacter* chromosomes including: (1) *Candidatus Campylobacter infans* str. 19S00001 (CP049075.1), (2) *C. avium* str. LMG 24591 (CP022347.1), *C. canadensis* str. LMG 24001 (CP035946.1), *C. coli* str. 14983A (CP017025.1), *C. coli* plasmid pCC14983A-1 (CP017026.1), *C. concisus* str. ATCC 33237 (CP012541.1), *C. corcagiensis* str. LMG 27932 (CP053842.1), *C. curvus* str. ATCC 35224 (CP053826.1), *C. fetus* str. NCTC 10354 (CP043435.1), *C. gracilis* str. ATCC 33236 (CP012196.1), *C. helveticus* str. ATCC 51209 (CP020478.1), *C. hepaticus* str. HV10 (CP031611.1), *C. hominis* str. ATCC BAA-381 (CP000776.1), *C. hyointestinalis* str. CHY5 (CP053828.1), *C. iguaniorum* str. RM11343 (CP015577.1), *C. insulaenigrae* str. NCTC 12927 (CP007770.1), *C. jejuni* str. NCTC 11168 (AL111168.1), *C. lanienae* str. NCTC 13004 (CP015578.1), *C. lari* str. RM2100 (CP000932.1), *C. mucosalis* str. ATCC 43264 (CP053831.1), *C. pinnipediorum* str. RM17261 (CP012547.1), *C. rectus* str. ATCC 33238 (CP012543.1), *C. showae* str. ATCC 51146 (CP012544.1), *C. sputorum* str. LMG 7795 (CP043427.1), *C. subantarcticus* str. LMG 24377 (CP007773.1), *C. upsaliensis* str. NCTC 11541 (LR134372.1), *C. ureolyticus* str. RIGS 9880 (CP012195.1), *C. volucrus* str. LMG 24380 (CP043428.1), and *C. vulpis* str. 251/13 (CP041617). As described for the *C. infans* analysis, Geneious Mapper was run with low sensitivity settings, which only map reads with <10% mismatch between reads and references. When <50 reads mapped to a particular genome in a sample, all of these reads were analyzed via BLASTn of NCBI nr/nt database to determine that the reads matched only a single species at >95% identity.

***BLASTn analysis of the* cadF *gene in a global dataset*.** Nucleic acid diagnostics in global multisite studies including MALED and GEMS have historically used custom array cards for the detection of multiple enteropathogens [16]. Among these, the detection of *C. jejuni* and *C. coli* has been based on the utilization of the following primers and probe targeting the *cadF* gene: Fw: 5'- CTG CTA AAC CAT AGA AAT AAA ATT TCT CAC -3', Rv: 5'- CTT TGA AGG TAA TTT AGA TAT GGA TAA TCG -3', and probe: 5'- CAT TTT GAC GAT TTT TGG CTT GA -3'. These targets were selected using standard genomic material including *Campylobacter coli* ATCC 43473, *Campylobacter jejuni* ATCC 33291. We performed a BLASTn analysis of the qPCR *cadF* forward and reverse primers using over 3,000 publicly available *Campylobacter* genomes in the PubMLST database.

## Results

Stool samples from 215 children with diarrhea as well as 50 randomly selected stool samples from children in a diarrhea free interval under the age of two were examined in this study. Among patients with diarrhea, 48% (103/215) were female and 52% were male (112/215). The

median age of symptomatic children was 14 months (IQR: 9 months– 19 months). Among asymptomatic patients, 54% (27/50) were female and 46% were male (23/50). The median age of asymptomatic children was 7.5 months (IQR: 5 months– 13 months).

## qPCR

Fecal samples were queried for *Campylobacter* spp., *C. jejuni*, *C. coli*, *C. infans*, and *Shigella* spp. using qPCR. The qPCR results (**Table 2**) indicated that overall, 55.1% (146/265) of fecal samples were positive for *Campylobacter* 16S rRNA. Of these, 35.6% (52/146) were positive for *C. jejuni* and/or *C. coli*, 13.0% (19/146) were positive for *C. infans*, and 51.4% (75/146) were negative for *C. jejuni*, *C. coli*, and *C. infans*.

Sixty percent (131/215) of the diarrheal samples were positive for *Campylobacter* spp. and approximately 23% (49/215) positive for *C. jejuni* /*C. coli*. All *C. jejuni*/ *C. coli* positive samples were positive for the 16S rRNA gene. The results from stool samples from asymptomatic children showed 30% (15/50) were positive for *Campylobacter* species and 6% (3/50) positive for *C. jejuni* / *C. coli*. *C. infans* was detected in 7.9% (17/215) symptomatic samples and 4.0% (2/50) asymptomatic samples. All *C. infans* positive samples were positive for the 16S rRNA gene. A single sample among diarrheal samples was positive for both the *C. jejuni*/ *C. coli* gene and *C. infans* by qPCR. Non-*C. jejuni* / *C. coli* and non-*C. infans* positive samples were identified in 30.2% (65/215) of symptomatic fecal samples and 20% (10/50) of asymptomatic fecal samples. *C. infans* explained 20.7% (17/82) of the diagnostic gap in samples from children with diarrhea and 16.7% (2/12) of the gap in children who had asymptomatic carriage.

Diarrhea was not associated with the presence of *C. infans* (OR: 2.06; 95% CI (0.46–9.22)). In comparison, diarrhea was significantly associated with the presence of *Campylobacter* spp. (OR: 3.64; 95% CI:1.87–7.07), *C. jejuni* / *C. coli* (OR: 4.62; 95% CI:1.38–15.51) and *Shigella* spp. (OR: 6.35; 95% CI: 1.48–27.14). (**Table 2**). The difference in the quantity of each qPCR target detected per gram of feces was not statistically different between symptomatic and asymptomatic fecal samples for *C. jejuni*, *C. coli*, or *C. infans* (**S1 Fig**).

DNA from twelve saliva samples associated with *C. infans* positive stools and twelve saliva samples associated with *C. infans* negative stools were further queried for *Campylobacter* spp., *C. jejuni*, *C. coli*, *C. infans* and *Shigella* spp. using qPCR. Four saliva samples were positive for *Campylobacter* spp. However, *C. jejuni*, *C. coli*, and *C. infans* were not detected in any saliva samples.

## Metagenomics

The *C. infans* qPCR assay identified 19/265 (7.2%) samples with a Ct value less than 38. The results of qPCR for *C. infans* were validated for 14 stool samples using shotgun metagenomic sequencing. The Ct values for *C. infans* in these 14 samples ranged from 37.16–19.97, which is approximately $10^5$–$10^{10}$ gene copies/g of feces. The metagenomic sequence reads were mapped

**Table 2. Detection of *Campylobacter* spp., *C. jejuni*, *C. coli*, *Candidatus Campylobacter infans* and *Shigella* spp. in symptomatic and asymptomatic fecal samples from children in the Peruvian Amazon using a multiplex qPCR.**

| Campylobacter marker | Symptomatic (N = 215) | Asymptomatic (N = 50) | Odds Ratios (95% Confidence Interval) |
|---|---|---|---|
| | % (n/N) | % (n/N) | |
| *Campylobacter* spp. (16S) | 60.9 (131/215) | 30.0 (15/50) | **3.64 (1.87–7.07)** |
| *C. jejuni* / *C. coli* (cadF) | 22.8 (49/215) | 6.0 (3/50) | **4.62 (1.38–15.51)** |
| *C. infans* (lpxA) | 7.9 (17/215) | 4.0 (2/50) | 2.06 (0.46–9.22) |
| *Shigella* spp. (ipaH) | 20.9 (45/215) | 4.0 (2/50) | **6.35 (1.49–27.14)** |

**Table 3. Total number of reads and number of reads detected for *Campylobacter infans*, *Campylobacter jejuni*, *Campylobacter coli*, *Campylobacter concisus*, *Campylobacter upsaliensis*, *Campylobacter helveticus*, *Campylobacter gracilis* and *Campylobacter hominis* from 14 stools that were PCR positive using the species-specific *C. infans* assay.**

| Sample | Total reads | C. infans | C. jejuni | C. coli | C. upsaliensis | C. concisus | C. gracilis | C. hominis | C. infans qPCR Cycle Threshold | log(10) Gene Copy Number/ gram of feces |
|---|---|---|---|---|---|---|---|---|---|---|
| 2854 | 10,720,574 | 69,948 | 7,235 | 332 | - | 2 | - | - | 25.56 | 9.36 |
| 2637 | 2,899,594 | 51,897 | - | - | 440 | 24 | - | - | 20.17 | 10.94 |
| 2050 | 2,618,280 | 36,829 | - | - | - | 2 | - | 4 | 23.63 | 9.93 |
| 352 | 2,789,880 | 35,540 | - | 13 | - | 12 | - | - | 23.08 | 10.09 |
| 982 | 3,942,970 | 32,000 | - | - | - | 6 | - | - | 19.97 | 11.00 |
| 2678 | 1,805,724 | 30,161 | 537 | 39 | - | 157 | - | - | 20.72 | 10.78 |
| 1842 | 1,553,388 | 2,532 | - | - | - | - | - | - | 23.75 | 9.89 |
| 1497 | 2,734,946 | 2,498 | - | - | - | | - | - | 25.65 | 9.34 |
| 320 | 1,976,662 | 1,775 | 81 | 6 | - | - | - | - | 26.29 | 9.15 |
| 1211 | 1,545,940 | 243 | - | - | - | 15 | - | - | 28.35 | 8.55 |
| 784 | 2,168,634 | 40 | - | - | - | 9 | 2 | - | 30.06 | 8.05 |
| 2706 | 392,074 | 26 | - | - | - | - | - | - | 27.37 | 8.83 |
| 2254 | 2,183,246 | 2 | 74 | 169 | - | - | - | - | 34.23 | 6.83 |
| 2125 | 1,287,704 | 0 | - | - | - | - | - | - | 37.16 | 5.97 |

Reads mapped to 27 *Campylobacter* genomes at >90% identity. Reads mapping to rRNA, AMR genes, IS elements/transposon were not included in counts except for the 3 highest *C. infans* read counts. 13/14 samples had *C. infans* specific reads. The lowest count was 2 (One set of paired reads that matched only *C. infans* at >98% in BLASTn of NCBI nr/nt database).

against the genomes of *C. infans* and 27 other *Campylobacter* species. *C. infans* sequencing reads were identified in 13 of the 14 samples using shotgun metagenomic sequencing (**Table 3**). All samples with a Ct value below 26.3 (>$10^9$ gene copies/g of feces) had more than 1,000 reads that mapped to the *C. infans* genome. One sample that had a Ct value of 34.23 (<$10^6$ gene copies/g of feces) had only 1 set of paired reads that mapped to the *C. infans* genome, and the sample with the highest Ct value of 37.16 (<$10^5$ gene copies/g of feces) had no reads. The number of reads matching to *C. infans* correlated with the quantity of *C. infans* calculated using the qPCR standard curve ($r(12) = 0.75$, *p-value* <0.05)

Shotgun metagenomic analysis identified reads for several *Campylobacter* species and showed many samples possessed multiple *Campylobacter* species (**Table 3**). Eight *C. infans* positive samples were co-infected with at least one non-*C. jejuni* / *C. coli* additional *Campylobacter* species identified from the metagenomic reference assemblies including 7 samples with *C. concisus*, one with *C. upsaliensis*, one with *C. gracilis* and one with *C. hominis*.

Despite *C. jejuni* and *C. coli* being scored as absent by qPCR (*cadF*) for 13 of the 14 samples, metagenomic analysis also identified four samples with *C. jejuni* and five with *C. coli*. This suggests that the *cadF* qPCR assay has a lower sensitivity than previously recognized. The qPCR *cadF* primers and probes were designed from a limited number of *cadF* sequences from *C. jejuni* and *C. coli* strains isolated in the US and/or Europe [4,17]. To determine the sequence diversity in the primer binding sites, we performed BLASTn analysis of the qPCR *cadF* forward and reverse primers (**Table 1**) against 3,076 *C. jejuni* and *C. coli* whole genome sequences (WGS) in the pubMLST database. There were 41% (1254/3076) of the strains that did not have 100% identity at one or both primer binding sites (**S2 Table**). Although most differences were only a single mismatch in a single primer, it is possible that the presence of multiple strains of *C. jejuni* and *C. coli* in a sample with differing mismatches may prevent proper amplification at a single *cadF* locus. It is also possible that *Campylobacter jejuni* and *Campylobacter coli* in

LMIC may exhibit additional sequence variation in and around the cadF gene as more than 90% of the publicly available whole genomes on *Campylobacter* global genome databases are derived from the United States and Europe. Despite the low number of metagenomic reads that mapped to *C. jejuni* and *C. coli*, four reads in two samples mapped to the *cadF* gene. One read from sample 2678 included the cadF-Fw primer-binding site with no mismatches. Unfortunately, this fact does not provide a reason for the negative *cadF* qPCR result.

## Discussion

There has been a consistent gap in detection and definitive identification of *Campylobacter* species in the clinical setting. To some extent this is a consequence of an inability to culture *Campylobacters* due to their fastidious nature. The acceptance of culture independent diagnostics (CIDT), such as ELISA and PCR, have led to some improvement in detection of *Campylobacter* from clinical samples. In fact, since 2013, *Campylobacter* has re-established as the leading cause of foodborne disease in the United States [18,19] and expanded its role as a major health issue worldwide. Despite the detection improvement of CIDT, detection gaps still exist due to the inability to identify certain *Campylobacters* due to the high variability of antigens targeted by ELISA, DNA sequence variation between species, and/or DNA variation within certain species at PCR primer/probe binding sites. The detection gap is especially pronounced in LMIC settings where culturing requirements including special gas mixtures are not easily obtainable and CIDT modifications can take months to develop and validate.

In the Peruvian Amazon, *Campylobacter* infections are endemic and have been detected using CIDT and culture-dependent genomic sequencing. Specifically, qPCR-based detection for *Campylobacter* species uses a specific, broadly reactive 16S rRNA gene target and for *C. jejuni* / *C. coli* uses a *cadF* gene target. In many cases, the qPCR results are validated by culture-dependent sequencing of *C. jejuni* and/or *C. coli* [20]. However, there are many samples that are positive for *Campylobacter* species (16S rRNA) while negative for *C. jejuni* / *C. coli* (*cadF*). Elimination of the other *Campylobacters*, as defined as *Campylobacter* 16S rRNA positive but *cadF* negative samples, would eliminate 24.9 percent of diarrhea cases in Peru, almost twice the attributable fraction of *C. jejuni* and *C. coli* [8]. Recently, a probable new species, *C. infans* was identified among infant stool samples from the GEMS study [12] and among patient samples in the Peruvian Amazon by metagenomic sequencing [21]. From these data, we modified the multiplex qPCR assay for detecting *Campylobacter* species and *C. jejuni* / *C. coli* to also detect *C. infans* using the *lpxA* gene.

The application of this multiplex qPCR assay was completed using a designed approach to include samples from health care attended cases of diarrhea as well as community-based controls. This study found the *C. infans* qPCR assay identified 19/265 (7.2%) samples with a Ct value less than 38, whereas shotgun metagenomic analysis of 14 of these samples, identified sequence reads that mapped to the *C. infans* genome in 13/14 (92.9%) samples. The single sample that had no reads had a Ct value of 37.16 and may indicate that a reduction of the positive Ct value is necessary for detection using shotgun metagenomics.

The validation of the *C. infans* qPCR assay using shotgun metagenomic sequencing not only demonstrated the presence of *C. infans* in positive samples but also demonstrated the presence of additional *Campylobacter* species that were not part of the qPCR assay. While 35.6% of the samples positive for *Campylobacter* species by qPCR were identified as positive for *C. jejuni* / *C. coli* in both the symptomatic and asymptomatic samples, the *C. infans* qPCR assay only increased the identification of the *Campylobacter* species to 48.6% of the samples. *C. infans* explained 20.2% of the diagnostic gap in samples from children with diarrhea and 16.7% of the gap in children who had asymptomatic carriage. Thus a large proportion of 16S

rRNA gene positive samples were not attributed to either *C. jejuni*, *C. coli* or *C. infans*. This leaves a rather large qPCR diagnostic gap between 16S rRNA positive *Campylobacter* samples and specific species identification. Here, the shotgun metagenomic sequencing of *C. infans* qPCR positive samples identified additional *Campylobacter* species within these samples including *C. jejuni*, *C. coli*, *C. concisus*, *C. gracilis*, *C. hominis*, and *C. upsaliensis*. In our initial shotgun metagenomic study to examine samples that were qPCR positive for *Campylobacter* species but negative for *C. jejuni* / *C. coli*, we also identified *C. jejuni*, *C. coli*, *C. concisus*, *C. upsaliensis*, *C. helveticus and C. curvus*. It is possible that new qPCR assays should be developed or validated in highly endemic settings to account for these other *Campylobacter* species. Despite previous studies evaluating the co-occurrence of multiple *Campylobacter* species in the human gut using shotgun metagenomics [9,11], these have yet to report a prevalence of *C. infans* in fecal samples.

The presence of *C. jejuni* / *C. coli* in several of these samples suggests that the *C. jejuni* / *C. coli cadF* qPCR assay may miss a subset of *C. jejuni* / *C. coli* strains. In fact, we also identified these two species by shotgun metagenomics in numerous similar stool samples that were determined to be positive for *Campylobacter* species but negative for *C. jejuni* / *C. coli* by qPCR [21]. BLASTn analysis of the primer binding sites for the *cadF* product in over 3,000 *C. jejuni* / *C. coli* genomes demonstrated that approximately 40% of the strains had at least one mismatch.

The *cadF* gene is considered a virulence factor and core genomic feature, but virulence profiling studies of *C. jejuni* /*C. coli* strains including several from South America report the absence of the *cadF* gene within *C. jejuni* / *C. coli* strains using PCR based detection [22–24]. Furthermore, BLASTn analysis of WGS contigs at 90% identity level over 90% of the gene length can also result in failure to detect the *cadF* gene, yet the presence of the *cadF* gene in all complete genomes of *C. jejuni* / *C. coli* supports its current identity as a core gene. Therefore, it is less likely that the shotgun metagenomics analysis is identifying *C. jejuni* / *C. coli* that are *cadF* negative, but PCR-based assays can occasionally fail to detect it in certain stool samples. Further shotgun metagenomic analysis of samples that are *Campylobacter* positive by 16S rRNA qPCR but *C. jejuni* / *C. coli* negative by *cadF* qPCR will be required to clarify this point.

As mentioned above, *C. concisus* was identified in 7 of the 13 samples in which *C. infans* was confirmed by shotgun metagenomic sequencing. *C. concisus* is an anaerobic or microaerophilic *Campylobacter* species that has been predominantly isolated from the human oral cavity [25,26]. The presence of oral bacteria, such as *C. concisus*, in the infant gut, has been previously described as microbiome decompartmentalization of the gastrointestinal tract and is associated with childhood stunting [27]. As a result of these facts and due to the many co-infections of *C. concisus* and *C. infans*, we hypothesized that *C. infans* may also be an oral bacterium. Therefore, we performed qPCR in 24 randomly selected saliva samples associated with *C. infans* positive and *C. infans* negative fecal samples. Although this assay identified *Campylobacter* spp. in four saliva samples, this assay failed to identify *C. infans* in any saliva samples. Further studies are still needed to rule out the focalization of *C. infans* in the human gastrointestinal tract.

Based on the qPCR findings of symptomatic versus asymptomatic samples, certain types of infections are more likely to be associated with diarrhea. The odds ratios for *C. jejuni* / *C. coli* infections and *Shigella* infections demonstrate an association with diarrhea. However, in this study, the odds ratio for *C. infans* infections suggest that *C. infans* infection is not associated with diarrhea. Overall, our findings delineate *C. infans* as having a relatively low prevalence and no important role in the etiology of diarrheal illness in this LMIC population.

## Conclusion

"*Candidatus* Campylobacter infans" was detected by CIDT in symptomatic and asymptomatic fecal samples from children in the Peruvian Amazon. Analysis determined that the presence of this potentially new species of *Campylobacter* was not associated with diarrhea in this population. Twenty-nine percent of *Campylobacter* qPCR positive fecal samples were not attributed to *C. jejuni*, *C. coli* or *C. infans*. Shotgun metagenomic analysis demonstrated the presence of other *Campylobacter* species and also suggested that qPCR detection of *C. jejuni* and/or *C. coli* using *cadF* failed to detect all samples that were positive for *C. jejuni* and/or *C. coli*. Further studies are needed to determine if the targeted section of the *cadF* gene that is widely used for the detection of *C. jejuni* and/or *C. coli* can be improved for the detection of strains present in LMICs.

## Supporting information

**S1 Table. gBlock Gene Fragments sequences used as positive controls.**
(DOCX)

**S2 Table. Highlighted Campylobacter isolates possess at least 1 mismatch at the forward or reverse primer binding sites.**
(XLSX)

**S1 Fig. Gene Copy Number (Log10) of qPCR Targets Per Gram of Feces in Symptomatic and Asymptomatic Fecal Samples.** Box and whiskers blot showing no statistical differences in the quantity of (Log10) of each qPCR target between symptomatic and asymptomatic samples. Quantitative data shown below:
(DOCX)

**S1 Data. Demographic, specimen type and qPCR data of individual samples.**
(XLSX)

## Author Contributions

**Conceptualization:** Francesca Schiaffino, Maribel Paredes Olortegui, Pablo Peñataro Yori, Kerry K. Cooper, Craig T. Parker, Margaret N. Kosek.

**Data curation:** Paul F. Garcia Bardales, Steven Huynh, Maribel Paredes Olortegui, Pablo Peñataro Yori, Tackeshy Pinedo Vasquez, Katia Manzanares Villanueva, Greisi E. Curico Huansi, Wagner V. Shapiama Lopez, Kerry K. Cooper, Craig T. Parker, Margaret N. Kosek.

**Formal analysis:** Paul F. Garcia Bardales, Francesca Schiaffino, Steven Huynh, Kerry K. Cooper, Craig T. Parker, Margaret N. Kosek.

**Funding acquisition:** Maribel Paredes Olortegui, Pablo Peñataro Yori, Kerry K. Cooper, Craig T. Parker, Margaret N. Kosek.

**Investigation:** Paul F. Garcia Bardales, Francesca Schiaffino, Maribel Paredes Olortegui, Pablo Peñataro Yori, Tackeshy Pinedo Vasquez, Katia Manzanares Villanueva, Greisi E. Curico Huansi, Wagner V. Shapiama Lopez, Kerry K. Cooper, Craig T. Parker, Margaret N. Kosek.

**Methodology:** Paul F. Garcia Bardales, Francesca Schiaffino, Steven Huynh, Maribel Paredes Olortegui, Pablo Peñataro Yori, Kerry K. Cooper, Craig T. Parker, Margaret N. Kosek.

**Project administration:** Margaret N. Kosek.

**Resources:** Pablo Peñataro Yori, Kerry K. Cooper, Margaret N. Kosek.

**Software:** Kerry K. Cooper, Craig T. Parker.

**Supervision:** Maribel Paredes Olortegui, Pablo Peñataro Yori, Kerry K. Cooper, Craig T. Parker, Margaret N. Kosek.

**Writing – original draft:** Paul F. Garcia Bardales, Francesca Schiaffino, Maribel Paredes Olortegui, Kerry K. Cooper, Craig T. Parker, Margaret N. Kosek.

**Writing – review & editing:** Paul F. Garcia Bardales, Francesca Schiaffino, Steven Huynh, Maribel Paredes Olortegui, Pablo Peñataro Yori, Tackeshy Pinedo Vasquez, Katia Manzanares Villanueva, Greisi E. Curico Huansi, Wagner V. Shapiama Lopez, Kerry K. Cooper, Craig T. Parker, Margaret N. Kosek.

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
