## [Decision Letter · Decision Letter 0]

14 Sep 2022

Dear Dr Kosek,

Thank you very much for submitting your manuscript "Prevalence of “Candidatus Campylobacter infans” in fecal samples of children under the age of 2 in Peru" for consideration at PLOS Neglected Tropical Diseases. As with all papers reviewed by the journal, your manuscript was reviewed by members of the editorial board and by several independent reviewers. The reviewers appreciated the attention to an important topic. Based on the reviews, we are likely to accept this manuscript for publication, providing that you modify the manuscript according to the review recommendations. 

Sincerely,

Paul J. Brindley, PhD

Editor-in-Chief

Abiola Senok

Academic Editor

Reviewer's Responses to Questions

**Key Review Criteria Required for Acceptance?**

**Methods**

-Are the objectives of the study clearly articulated with a clear testable hypothesis stated?

-Is the study design appropriate to address the stated objectives?

-Is the population clearly described and appropriate for the hypothesis being tested?

-Is the sample size sufficient to ensure adequate power to address the hypothesis being tested?

-Were correct statistical analysis used to support conclusions?

-Are there concerns about ethical or regulatory requirements being met?

Reviewer #1: Materials and Methods

With regard to data and result reporting please provide full details when reporting proportions and be more clear in presenting how much out of how much and what. Also it is not that odds of diarrhea are associated with presence of Campylobacter, it is diarrhea itself, and that relationship is expressed through odds ratio. Please be careful in writing of results. 

Line 127 – what about adjustment for other gene targets?

Line 143 – why were these changes to the protocol needed?

Reviewer #2: Adequate methods

**Results**

-Does the analysis presented match the analysis plan?

-Are the results clearly and completely presented?

-Are the figures (Tables, Images) of sufficient quality for clarity?

Reviewer #1: Results

Line 184 – I guess it should be median and not mean.

Line 197 – were not significantly higher sounds like a one-tail test was performed, and if it was not why not write not significantly different (includes both higher and lower).

It wold be helpful in presenting Campylobacter +ve, C. jejuni/coli +ve, C. infans +ve and other species to provide a full cross-tabulated details to follow what are the actual results. What I mean, is data, for instance, how many 16S qPCR +ve samples had +ve results od species-specific qPCRs? Were there any species-specific +ve results that were -ve by 16S qPCR etc. Also, how was actually the prevalence of other, untested, Campylobacter species calculated (provide numerators and denominators).

Line 221 – how correlated? were correlation tests used?

Line 234 – were all mismatches only single mismatches per primer but sometimes single on both primers? You place a lot of emphasis on this to support differences in sensitivity of PCR, this could have easily been checked with degenerate primers in equimolar amounts. Why wasn’t it tried? Why didn’t you test that by mapping your metagenomic reads to cadF? This study was a perfect opportunity to test other primers (hipO, mapA. ceuE.., or modify them to support these claims (at least on smaller subsets).

Discussion

Line 243 – this is wrong. Campylobacter was already recognised as the leading cause before culture-independent methods were being used. You could check history of Campylobacter since late 60’s and early 70’s when finally a method for culturing was devised in Belgium. Everyone in the world started then using it and reporting it as the leading cause…

Line 249 – why would this explain in LMIC? If culture is harder to do then molecular methods are used instead? That would imply better sensitivity so no gap….please make clearer.

Please do not write results in the discussion. Use it to discuss the data not repeating of results section.

Line 284 – yes, this study was a good opportunity for that!

In general, the discussion could have been more thorough. There are many studies using variety of culture and culture-independent methods. Many gene targets exist, many other species are also emerging and have various results between studies (C. upsaliensis especially…)…

Tables

Table 1 – Source not fuente. Also provide literature reference for primers, that is more common form.

Table 3 – was 98% cutoff used for other species too?

Supplementary figure 1 – the boxplots for C. infans show very clearly a different mean location between asymptomatic and symptomatic children. It is very unusual that t test would return the p value of 0.07. Please check data and test implementation and re-run. If this is not different, I’d like to see raw data and test it myself.

Reviewer #2: The result section is adequate

**Conclusions**

-Are the conclusions supported by the data presented?

-Are the limitations of analysis clearly described?

-Do the authors discuss how these data can be helpful to advance our understanding of the topic under study?

-Is public health relevance addressed?

Reviewer #1: Yes, the conclusions follow the data presented and are appropriately described.

Reviewer #2: Although the overall number of Campylobacter strains included in the study is adequate, I think the authors cannot conclude that this potential new species (C. infans) has no role in the etiology of diarrhea illness in children. I think more studies are needed in other sites in order to determine its real role.

**Editorial and Data Presentation Modifications?**

Reviewer #1: With regard to data and result reporting please provide full details when reporting proportions and be more clear in presenting how much out of how much and what. Also it is not that odds of diarrhea are associated with presence of Campylobacter, it is diarrhea itself, and that relationship is expressed through odds ratio. Please be careful in writing of results.

Reviewer #2: None

**Summary and General Comments**

Reviewer #1: Overall comments

The authors report a diagnostic cross-sectional study of Campylobacter spp.in children in Peru . Molecular techniques have been used for detection and the authors investigated primarily the role in gastrointestinal illness of an emerging Campylobacter species proposed as C. infans. The results showed it is not associated with symptomatic illness and that there is more unaccounted Campylobacter positive infections based on 16S rDNA PCR and metagenomic analyses that are negative by species-specific qPCR used in the study. This is a valuable study for epidemiology of GI pathogens in low income settings and will be of interest to the readership.

With regard to the language and writing, an overall remark is that writing style and format needs to be consistent e.g., writing of numbers at the beginning of the sentence, usage of symbols (%). There are a lot of instances of unnecessary long and convoluted sentences that could be shorter, clearer and more concise. I would suggest one native english speaker to improve the manuscript.

Specific comments

Keywords – perhaps to change Iquitos to Peru? Could be a better option in literature search.

Abstract

Line 30 – investigated instead of measured.

Lines 36-38 – rewrite clearer and shorter. e.g. C. infans presence was not significantly associated with symptomatic illness. Use space gained to perhaps add data on the sample population (age, gender?) or add data on other Campylobacter species detected…

Introduction

Line 50 – so, where is the disease highest? And that, as written, is not evidence of the discrepancy between culture and culture-independent diagnostic methods. Rewrite as it is confusing.

Line 53 – where is second reason written? Is it the emerging Campylobacter species?

Line 68 – which unidentified species, mapped how and to what? Then, similarity in what? Provide details.

Line 67 and 73 – do not report % only data, provide out of how many sequences, samples…

Reviewer #2: This is a interesting study on Campylobacter, evaluating the relevance of a potential new Campylobacter species (C. infans) in children from the Paruvian Amazon. The study includes qPCR and metagenomics studies. 

Minor comments

1. Although the overall number of Campylobacter strains included in the study is adequate, I think the authors cannot conclude that this potential new species (C. infans) has no role in the etiology of diarrhea illness in children. I think more studies are needed in other sites in order to determine its real role. 

2. In the method and results sections, it is not clear why the authors performed studies on saliva samples. It is on the discussion section that the authors explain the reason of these studies (potential presence of C. infans in the oral cavity similar to C. concisus). This should be mentioned early on the text. 

3. A very relevant result of the study is the evaluation of the cadF gene for the detection of C. jejuni and C. coli. This should be part of the title.

PLOS authors have the option to publish the peer review history of their article (what does this mean?). If published, this will include your full peer review and any attached files.

Reviewer #1: No

Reviewer #2: No

Figure Files:

Data Requirements:

Reproducibility:

References

---

## [Editor Report · Decision Letter 1]

5 Oct 2022

Dear Dr Kosek,

We are pleased to inform you that your manuscript '“Candidatus Campylobacter infans” detection is not associated with diarrhea in children under the age of 2 in Peru' has been provisionally accepted for publication in PLOS Neglected Tropical Diseases.

Best regards,

Paul J. Brindley, PhD

Editor-in-Chief

---

## [Editor Report · Acceptance letter]

13 Oct 2022

Dear Dr Kosek,

We are delighted to inform you that your manuscript, "“Candidatus Campylobacter infans” detection is not associated with diarrhea in children under the age of 2 in Peru," has been formally accepted for publication in PLOS Neglected Tropical Diseases.

Best regards,

Shaden Kamhawi

co-Editor-in-Chief

Paul Brindley

co-Editor-in-Chief
